# Corrosion Behavior of Magnesium Potassium Phosphate Cement under Wet–Dry Cycle and Sulfate Attack

**DOI:** 10.3390/ma16031101

**Published:** 2023-01-27

**Authors:** Linlin Chong, Jianming Yang, Jin Chang, Ailifeila Aierken, Hongxia Liu, Chaohuan Liang, Dongyong Tan

**Affiliations:** 1College of Civil Engineering, Changsha University, Changsha 410022, China; 2Innovation Center for Environmental Ecological and Green Building Materials of CCSU, Changsha University, Changsha 410022, China; 3School of Civil Engineering, San Jiang University, Nanjing 210012, China

**Keywords:** magnesium potassium phosphate cement (MKPC), sulfate attack, dry–wet cycles, strength change, volume stability

## Abstract

This paper investigated the influence of dry–wet cycles and sulfate attack on the performance of magnesium potassium phosphate cement (MKPC) as well as the effect of waterglass on MKPC. X-ray diffraction (XRD), TG-DTG, and scanning electron microscopy (SEM-EDS) were used to examine the phase composition and microstructure of MKPC. The results showed that the flexural and compressive strength of an MKPC paste increased initially and subsequently decreased in different erosion environments. The final strength of the M0 paste exposed to the SK-II environment was the highest, while that of the M0 paste exposed to the DW-II environment was the lowest. The final volume expansion value of MKPC specimens under four corrosion conditions decreased in the following order: DW-II, M0 > SK-II, M0 > DW-II, M1 > SK-I, M0 > DW-I, M0. Compared to the full-soaking environment, the dry–wet cycles accelerated sulfate erosion and the appearance of damages in the macro and micro structure of the MKPC paste. With the increase in the number of the dry and wet cycles, more intrinsic micro-cracks were observed, and the dissolution of hydration products was accelerated. Under the same number of dry–wet cycles, the strength test and volume stability test showed that the durability in a Na_2_SO_4_ solution of the MKPC paste prepared with 2% waterglass (M1) was superior to that of the original M0 cement. The micro analysis indicated that waterglass can improve the compactness of the microstructure of MPC and prevent the dissolution of struvite-K.

## 1. Introduction

As one of the most widely used construction materials, concrete is increasingly being used in difficult environments such as salt lakes, salty soil, collapsible soil, as well as in maritime engineering [1]. However, the concrete structures used under such conditions suffer severely from long-term corrosion caused by sulfate attack, which severely affects the normal performance of buildings [2,3,4]. Furthermore, differences in river and groundwater levels with the changes in season, as well as the splash zone, result in dry–wet cycles. Sulfates have a more severe effect on concrete structures when combined with wet–dry cycles. The combined effects of wet–dry cycles and sulfate pose a greater risk of damage to concrete structures. Magnesium phosphate cement (MPC) has been promoted for the rapid repair of concrete structures in various corrosive environments in order to extend their service life [5]. MPCs have excellent properties as compared to other cementitious materials, such as enhanced adaptability to environmental temperature, fast hardening, high early strength, low shrinkage, better adhesion, etc. [6,7,8,9,10,11].

Magnesium phosphate cements (MPC) are a class of inorganic cementitious materials containing phosphates as their binder phase, which are formed through neutralization reactions occurring in the cement [6]. In recent years, more attention has been paid to MPC-based repair materials. Research mainly focuses on its preparation technology, modification mechanisms, and repair in the normal environment. With the development of research on MPC, some studies have investigated the water stability [12,13,14,15], salt stability (through long-term soaking in a salt solution) [16,17,18,19,20], salt freeze–thaw durability of MPC-based materials [4,21,22] and the passivation of steel [23,24,25,26]. Some researchers determined that MPC, unlike common concrete, has excellent sulfate resistance. For example, Li et al. [14] found that the strength of MPC in a Na_2_SO_4_ solution was higher than that in water and in a NaCl solution. Yang et al. [18,19] revealed that the strength of MPC pre air curing for 28 d was 93% of the original strength after immersion in seawater and even slightly increased after immersion in a Na_2_SO_4_ solution for 360 days. MPCs used as coating materials for concrete also showed good sulfate resistance [27,28]. As a repair material used in marine concrete construction, MPC is also subjected to sulfate attack in a dry–wet cycling environment. However, only one study described the durability of MPC-based materials during dry–wet cycles in water and salt solutions. Li [29] examined the degradation of MKPC and the effects of fly ash and quartz sand under the same corrosion environment. An MKPC paste with quartz sand showed poor durability when it was treated with dry–wet cycles. By adding fly ash to the MKPC paste, the authors observed a drop in compressive strength and mass. Clearly, more work in this respect is needed to better predict the performance of MPC exposed to marine conditions.

This study aimed to provide further information about the corrosion behavior of an MKPC paste subjected to the effects of sulfate attack and dry–wet cycles by focusing on changes in compressive strength and flexural strength, volume deformation, and microstructure changes. Plain MKPC pastes and a paste with 2% waterglass subjected to dry–wet cycle and sulfate attacks were tested for about one year. A plain paste was also subjected to water immersion and water dry–wet tests for comparison. The phase composition as well as the microstructure changes were evaluated at the end of the test period by means of SEM, EDS TG-DTG, and XRD. This study provides useful information for improving the durability of MPC-based materials and their application in the reinforcement of concrete structures subject to both salt attack and dry–wet cycles.

## 2. Materials and Methods

### 2.1. Raw Materials

Dead burnt magnesium was obtained from the Huan ren Dong fang hong Hydropower Factory for Station Magnesite (Liaoning province, China). The specific surface area of magnesium oxide was 230 m^2^/kg, its average particle size was 45.26 µm, and its chemical composition determined by X-ray fluorescence analysis is shown in Table 1.

Industrial-grade potassium dihydrogen phosphates (KH_2_PO_4_, PDP) were purchased from Georgia Legislature Chemical Company (Lianyungang, Jiangsu province, China). They consisted of colorless or white crystals with a main particle size of 40/380–60/250 (mesh/μm). A composite retarder (CR) [13] was used as a setting retarder. Waterglass with Baume degree of 39.2–40.2 and modulus of 3.2–3.4 was from Yixing Credible Chemical Co., Ltd.

### 2.2. Specimen Preparation

The mass ratio of alkali component (MgO) to acid component (KH_2_PO_4_) was set at 3 according to previous results of superior mechanical performance [4,13,30]. Meanwhile, to obtain samples with both good mechanical strength and good workability, the mass ratio of water to solid (*w*/*s*) was 0.115. The solid contained MgO, KH_2_PO_4_, and the composite retarder (CR). The content of the composite retarder was 12.0% by weight of MgO, and the added waterglass (by weight of the solid) was 2 wt.%. The MKPC mix proportions are summarized in Table 2. During casting, MgO powder, PDP, and CR were firstly dry-mixed at a low speed for 1 min, and then water was added and stirred rapidly for another 3 min. Then, the fresh MKPC paste was quickly cast into specimens with dimensions of 40 × 40 × 160 mm and 25 × 25 × 280 mm to test the compressive strength and volume deformation. All the molds were covered with plastic sheets, and the samples were cured for 5 h at room temperature and then demolded. They were then cured indoors at a temperature of 20 ± 5 °C and a relative humidity of 60 ± 5% for 28 days.

### 2.3. Environmental Exposure Conditions

After natural curing for 28 days, a batch of MKPC specimens were cured for a longer time under natural conditions and used as reference samples. The remaining samples were divided into four groups and subjected to four different exposure conditions (listed in Table 3) up to 360 days. Part of the specimens were subjected to a full-soaking (i.e., SK) environment, while the other specimens were exposed to a dry–wetting cycle (i.e., DW) environment. In both environments, tap water without sodium sulfate and a 5% Na_2_SO_4_ solution were selected as the media, which were replaced with fresh ones every 30 days. In the DW environment, the specimens were first immersed in water or the Na_2_SO_4_ solution for 23 ± 0.5 h (until reaching a full water state) and then dried at 60 °C at a vacuum-drying temperature for 47 ± 0.5 h (to reach a constant mass state). After drying, the specimens were cooled at room temperature for 2 h. Each dry–wet cycle lasted for 3 days; 120 dry–wet cycles were carried out on the specimens in total, which equated to 360 days of immersion.

### 2.4. Experimental Methods

#### 2.4.1. Fluidity and Setting Time

Based on ASTM C1437-2007 [31], NLD-3 jump tables were used to measure the fluidity of the fresh MKPC paste. The MKPC paste initial setting time was measured and recorded following ASTM C191a-2001 [32], using a Vicar apparatus.

#### 2.4.2. Mechanical Testing

According to ASTM C348-2008 [33] and ASTM C 349-2008 [34], three samples for each batch were used to evaluate the flexural strength by a WED-300 electronic universal testing machine at loading rates of 40 to 60 N/s, and six broken samples were then used for the compressive strength test at a loading rate from 2200 to 2600 N/s. The flexural strength and the compressive strength of the MKPC specimens over time were tested in a saturated-surface dry state. The residual strength ratios were calculated as Kf = Rfn/Rf0 and Kc = Rcn/Rc0, where Rf0 (Rc0) and Rfn (Rcn) are the flexural (compressive) strengths of the MKPC specimens (saturated-surface dry state) after 0 and t dry–wet cycles, respectively.

#### 2.4.3. Volume Stability

MKPC paste bars with a size of 25 × 25 × 280 mm were measured every two months by using a Digimatic outside micrometer. The saturated-surface-dry length (L0) at the beginning of the corrosion test was measured, as well as the length of the specimens (Lt) after t dry–wet cycles. The effective length of the sample is shown here based on 250 mm. The deformation rate (εn) was calculated based on JC/T603–2004 [35] as follows:(1)$$\varepsilon_{t}={(L}_{t}{-L}_{0})/250\times 100\%$$

#### 2.4.4. Microanalysis

Following the strength tests, samples of the MKPC paste were collected from the broken specimens for microscopic analysis. To stop the hydration of the samples, they were submerged for at least 2 days in anhydrous alcohol. Then, they were dried to constant mass in a vacuum-drying chamber at 60 °C. Some parts were ground into powder and passed through a 45 μm sieve for TG and XRD analyses. The phase composition of the samples was determined using Rigaku Neo D/max RB X-ray diffractometers (Cu-Ka radiation in the 5° to 80° range with a step size of 0.02°, 40 kV, 100 mA). The powder samples were heated at 10 °C/min from 20 °C to 1100 °C using a NETZSCH STA 409 PC/PG thermal analyzer under the protection of N_2_. MKPC paste samples were analyzed using a quantitative environmental scanning electron microscope (Quanta-200, American FEI Company) to determine their morphologies and by energy-dispersive X-ray spectroscopy (EDS).

## 3. Results and Discussion

### 3.1. Strength Development

Figure 1 depicts the development of flexural strength and compressive strength in the MKPC paste under different corrosive environments. As shown in Figure 1, the flexural strength and compressive strength of both M0 and M1 increased first to a peak and then decreased as the erosion grew in the four corrosion conditions, while they increased continuously under air curing. After 40 dry–wet cycles, both M0 and M1 showed a higher compressive strength than the air-cured samples, possibly due to better paste hydration resulting from corrosion-induced water adsorption [13,18,36]. The M0 paste subjected to the DW environment at first had a higher strength than the MKPC specimen subjected to the SK environment under the same hydration time. After 120 dry–wet cycles, the final strengths of the samples were in the order AC, M0 > SK-II, M0 > DW-I, M0 > SK-I, M0 > DW-II, M1 > DW-II, M0. The flexural strength and compressive strength of M0 exposed to the SK-II environment were 12 MPa and 63.5 MPa, the highest among those of all the specimens except for that of the air-cured samples. In contrast, the M0 paste exposed to the DW-II environment had the lowest strength. This indicated that wet–dry cycling appeared to be beneficial for improving the water resistance of MKPC, but detrimental for its sulfate resistance. During sulfate attack, the M1 paste reached the peak strength after 40 dry–wet cycles, while the M0 paste reached the peak strength after 20 dry–wet cycles, and the peak strength of the M1 paste was greater than that of the M0 paste. Under the DW-II environmental conditions, this could be explained by a more dense structure and a reduction in the dissolution of hydration products due to the addition of waterglass, resulting in an increase in the flexural and compressive strength and durability of MKPC.

The main raw material in MKPC (i.e., dead burned MgO powder) is obtained by calcining magnesite (MgCO_3_) at above 1300 °C; so, the surfaces of the obtained particles are smooth and compact and have low water absorption [11,37]. Consequently, the water-to-cement ratio of the fresh MKPC paste necessary to reach the required consistency is low (Table 2), but it is not sufficient for a complete hydration of the MKPC paste. Therefore, there are still some unreacted phosphates in addition to the main hydration product MgKPO_4_·6H_2_O and many unreacted dead burned MgO particles in the hardened MKPC paste. Subsequent hydration in the MKPC paste is closely related to the environmental humidity [13,18,30]. For example, when an MKPC specimen is soaked in water or a salt solution, the infiltration of ambient water will lead to its further hydration. The newly generated MKP crystals will fill the micropores in the hardened MKPC paste, leading to an increase in the strength of the MKPC specimen [13,30]. This explains why the strengths of the MKPC specimens immersed in water and the 5% Na_2_SO_4_ solution increased first. Particularly, the strength of the MKPC specimen immersed in the 5% Na_2_SO_4_ solution was higher than that of the MKPC specimen immersed in water for the same period. This was due to the filling effect of some newly generated sulfate-containing crystals in addition to the MKP crystals [14,22].The process of vacuum-drying at 60 °C can accelerate the mutual penetration between unreacted acid and alkali components and promote the hydration process, so that more newly generated hydrates will fill the pores. This is the reason why the hardened MKPC paste subjected to dry–wet cycles with water and 5% Na_2_SO_4_ at first had higher strength than the MKPC specimen soaked in water and 5% Na_2_SO_4_ for the same hydration time. With the extension of corrosion, the hydration reaction in the hardened MKPC paste tends to stop, and the over-expansion and decomposition of the erosion products begin to play a leading role in the subsequent degradation of the MKPC specimens under sulfate and water attacks, as corrosion continues [18]. Compared to the M0 samples exposed to the SK-I environment, the M0 samples under dry–wet cycles with water were only intermittently in a water-rich environment, which decreased the hydrolysis and loss of the MKP in the specimens exposed to the DW-I environment and led to a higher strength. However, when the M0 samples were cured under the DW-II environment, the hardened MKPC paste absorbed the solution by capillary action. The salt crystallized as the water evaporated in the vacuum-heating stage. When the salt crystals in the capillaries absorbed water and converted to a state containing crystal water, a high crystallization pressure was produced. This could result in cracking and peeling of the surface of the hardened MKPC paste and corner damage, finally leading to a strength reduction in the MKPC specimens.

### 3.2. Strength Retention Coefficient

Figure 2 shows the strength retention coefficient of the MKPC paste under various corrosive conditions. Compared to the initial strength, the strength retention coefficient of the M0 paste under different corrosive environments increased first and reached a peak value at 60 dry–wet cycles. The strength residual rate of the specimens (M0) soaked in water and 5% Na_2_SO_4_ was 106.5% and 107.3% (flexural strength) and 113.3% and 117.5% (compressive strength). The strength residual rate of the specimens after 60 dry–wet cycles in water and 5% Na_2_SO_4_ was 108.9% and 102.4% (flexural strength) and 120.2% and 113.4% (compressive strength). After 120 dry–wet cycles, the strength residual rate of the specimens (M0) when soaked in water and 5% Na_2_SO_4_ was 90.2% and 97.56% (flexural strength) and 91.1% and 99.2% (compressive strength), the strength residual rate of the specimens after 120 dry–wet cycles in water and 5% Na_2_SO_4_ was 94.3% and 74.6% (flexural strength) and 97.2% and 79.9% (compressive strength). M0 soaked in the sulfate solution exhibited significantly better corrosion resistance than after soaking in water. It appears that M0 had a higher resistance to sulfate corrosion than to water corrosion, as previously reported [18]. However, the M0 paste under dry–wet cycles in the 5% Na_2_SO_4_ solution presented the lowest strength retention ratio in the whole corrosion test when the strength started to drop. This indicates that dry–wet damage was dominant when the samples were exposed to the combined action of sulfate attack and wet–dry cycling.

According to the results of M0 and M1, the increase in the strength of the M1 samples took longer than that of M0, and the strength residual rates of the M1 samples were 83.6% (flexural strength) and 89.8% (compressive strength). The strength residual rates of the M1 samples were more than 5% higher than those of M0 under DW-II conditions, indicating that adding waterglass to the MKPC paste would improve its resistance to dry and wet sulfate cycles. When waterglass is added to MKPC, H_3_SiO_4_^−^, H_2_SiO_4_^2−^ and HSiO_4_^3−^ will react with Mg^2+^ in the MKPC paste to form magnesium silicate gels [12], which can fill the pores and make the structure dense. When the hardened MKPC paste was immersed in water, its dense initial structure prevented external water from infiltrating it and the dissolution of MKP and it could also prevent the infiltration of the sulfate solution which in turn reduced the formation of sulfate-containing crystals and the destruction of salt crystals when subjected to dry–wet cycles with the 5% Na_2_SO_4_ solution.

### 3.3. Volume Stability

Figure 3 shows the evolution of the deformation of the MKPC specimens exposed to the four corrosive environments. All specimens exhibited expansion deformation, and the value increased as the number of dry–wet cycles increased. This can be attributed to the increased degree of hydration in environments with excess liquid phase [36]. Furthermore, the unreacted MgO and H_2_O formed Mg(OH)_2_, resulting in volume expansion [10,38]. Typically, M0 exhibited an expansion deformation pattern consistent with its corrosion age, except in DW-II corrosion conditions. The specimens swelled rapidly within 60 dry–wet cycle, and the expansion rate was moderate. Compared to M0 cured under SK-II environments, the expansion of M0 under DW-II conditions was significantly lower when the number of dry–wet cycles was limited to 40. The expansion tended to be more pronounced when the number of dry–wet cycles exceeded 80. Initially, the MKPC specimens subjected to dry–wet cycling were significantly less deformed than the MKPC specimens soaked in water or the sulfate solution. This can be explained by the fact that the MKPC specimens soaked in the corrosion solution were always water-saturated, while those subjected to the dry–wet cycles were in a drying state, which means they would not undergo deformation due to crystal expansion as a result of water absorption. A further increase in the number of the dry–wet cycles would result in an expansion of the MKPC specimen. This can be attributed to the crystallization pressure of salt resulting from the water-saturated crystallization of the sulfate salt during drying, as well as the newly formed hydrated product (MKP) and the newly generated sulfate-containing phases. The volume expansion rate of M0 soaked in the 5% Na_2_SO_4_ solution was much higher than that of M0 soaked in water throughout the whole process. This can be attributed to the formation of new sulfate-containing crystals such as MgK_2_(SO_4_)_2_·6H_2_O [19]. After 120 dry–wet cycles, the volume expansion rates were in the order DW-II, M0 (3.62‰) > SK-II, M0 (2.77‰) > DW-II, M1 (2.73‰) > SK-I, M0 (1.89‰) > DW-I, M0 (1.28‰).

The volume expansion rate of M1 was less than that of M0 under the same conditions, indicating that the addition of waterglass could obviously improve the volume stability of the MKPC paste. By combining M1 with some water glass, magnesium silicate gels formed and could fill the pores, making the structure dense. As a result, external water could not infiltrate to the structure, thereby reducing the formation of sulfate-containing crystals, as well as the expansion and deformation caused by the crystallization pressure.

### 3.4. XRD Analysis

Figure 4 shows the XRD patterns of the MKPC samples exposed to the four corrosion environments. The patterns revealed that MgKPO_4_6H_2_O (MKP) was the primary hydration product, and a significant amount of unreacted MgO was still present. The main characteristic peak positions of the MKPC samples soaked in water and subjected to 120 dry–wet cycles in water were basically the same, indicating that no extra new phases formed. Compared to the M0 samples soaked in water, the MKP samples soaked in the sulfate solution exhibited a stronger diffraction peak intensity, suggesting excellent crystallinity of the hydration products. There were two new main characteristic peaks corresponding to MgK_2_(SO_4_)_2_·6H_2_O and Na_2_SO_4_·10H_2_O in the MKPC sample subjected to dry–wet cycles in the 5% Na_2_SO_4_ solution. It appeared that the sulfate ion from the salt solution combined with the cation in the hardened MKPC paste to form a hydrate containing sulfate. Na_2_SO_4_·10H_2_O should be produced by water-saturated crystallization of sulfate in the pores during drying of the MKPC specimens. These new phases will increase the crystallization pressure of salt and cause volume expansion. Additionally, the main diffraction peak of MKP showed the weakest intensity, which could be due to its significant dissolution. These factors led to the worst structural degradation of the hardened MKPC specimens.

Compared to the M0 sample soaked in water, we observed more than one new main characteristic peak of MgSiO_3_ at 39.8–40.0 2θ° in the M1 sample soaked in water, which should be due to the new phase produced by the reaction of silicate ions and Mg^2+^ in waterglass. Compared with the M0 sample subjected to 120 dry–wet cycles in 5% Na_2_SO_4_, the intensity of the main diffraction peak of MKP in M1 was obviously higher than that in M0. In addition, the main characteristic peaks of Na_2_SO_4_·10H_2_O and MgK_2_(SO_4_)_2_·6H_2_O were missing in the M1 sample, which explains why the water-saturated crystallization of sulfate in the pores decreased during the drying process of the M1 specimen. The structural degradation of the M1 specimen caused by the volume expansion and the crystallization pressure of salt was relieved.

### 3.5. TG-DTG Analysis

Figure 5 exhibits the TG-DTG curves of MKPC after 360 days of immersion and 120 dry–wet cycles in water and the 5% Na_2_SO_4_ solution. At about 100 °C, the DTG curve illustrates a significant loss of mass due to the loss of crystalline water [39,40]. Moreover, an additional peak between 350–400 °C was observed when the MKPC paste was subjected to 120 dry–wet cycles in 5% Na_2_SO_4_, which could be due to the dehydration of the new sulfate-containing crystals; further study will be needed. Below 200 °C, the percentages of mass loss of the MPPC specimens were as follows: SK-II, M0 (11.56%) > DW-I, M0 (11.48%) > SK-I, M0 (11.43%) > DW-II, M1 (10.91%) > DW-II, M0 (9.92%). In the same corrosion conditions, the total mass loss below 200 °C in the M1 sample was obviously larger than that in M0, which confirmed that the hydrolysis and loss of MKP in M1 were more limited than that in M0, but the little mass loss at 300–400 °C in M1 was obviously less than that in M0.The lowest quantities of the hydration product MKP in the M0 samples cured under the DW-II environment indicated that hydrolysis and loss of MKP were enhanced due to the deterioration of the pore structure caused by the crystallization pressure of the salt, justifying the strength test results during this process.

### 3.6. SEM-EDS Analysis

Figure 6a–f shows the microstructure of the MKPC samples after 12 months of exposure to the SK and DW environments in water and sulfate solution. The EDS results are shown in Table 4. When the MKPC paste was exposed to the SK environment in water (Figure 6a), good columnar crystals appeared tightly packed together in an orderly manner. These crystals (area A) were verified as MKP on the basis of a molar ratio of Mg/P/K of approximately one. Some Na and CI elements were also detected in area B, probably derived from the composite retarder. In the M0 sample exposed to the SK-II environment, distinct morphologies such as rod-like and needle-like crystals of hydrates were observed (Figure 6b,c); these hydrates may be the products of struvite-K dissolution and precipitation. As shown in Table 4, the components of Area C were O, Mg, P, K, Na, CI, and S elements, with a molar ratio of P to K of approximately 1:1. Based on the results of XRD, these elements should be derived from the hydration products and the remaining magnesium oxide. Among them, the presence of the S element confirmed that a sulfate-containing phase was mixed with the hydration products. On the fracture surface of the M0 sample subjected to the DW-I environment (Figure 6d), well-formed and randomly oriented rod-like crystals shown to be MKP by the EDS results were observed. The crystal surface was smooth without visible corrosion marks. The combination of the hydration products such as MKP and the remaining MgO made the microstructure dense, which could also explain the good compressive strength and deformation stability. After 120 dry–wet cycles in the sulphate solution, the microstructure showed in Figure 6e of the M0 samples was very loose, more large rod-like crystals were observed—some of them broken—and many amorphous phases were distributed among the crystals and adhered to their surfaces, indicating the dissolution of the hydration products in the MKPC past with the formation of pores [13]. Based on the high S content in area D, it was evident that SO_4_^2−^ was involved in the hydration reaction, which resulted in new hydration products. The generation of Mg(OH)_2_ occurred as MgO reacted with H_2_O, resulting in a high internal stress that promoted the formation of intrinsic microcracks, further resulting in the decrease of strength and volume stability. These results are also in accordance to the XRD and TG-DTG results presented in Section 3.3 and Section 3.4.

Moreover, the M1 samples showed well-compacted structures in comparison to the M0 samples under DW-II conditions, with many orderly MKP crystals forming prismatic shapes on a smooth, defect-free crystal surface (Figure 6f). Some amorphous phases were present in the crystals and adhered to their surfaces. The addition of waterglass improved the internal structures of MKPC and avoided the dissolution of MKP. Therefore, the M1 samples had higher resistance to sulfate corrosion-coupled wet and dry cycles. A magnesium silicate hydrate gel would also be formed by waterglass and magnesium ions [18], these results were confirmed by EDS in area E, and concurred with previously presented XRD results. Although the crystal size of the M1 samples was bigger and the samples were well crystallized, the amount of crystals was less than in the M0 samples cured under the SK environment, as seen from the TG-DTG curves; therefore, the strength and volume stability were lower.

## 4. Conclusions

In this study, the corrosion behavior of magnesium potassium phosphate cement under wet–dry cycle and sulfate attack was investigated. Conclusions can be drawn as follows:After 120 dry–wet cycles, the final strengths of the MKPC paste were in the order AC, M0 > SK-II, M0 > DW-I, M0 > SK-I, M0 > DW-II, M1 > DW-II, M0. The dry–wet cycles in water caused less corrosion of the MKPC paste compared to soaking in water, and dry–wet damage played a dominant role when the samples were exposed to the combined action of 5% Na_2_SO_4_ sulfate and dry–wet cycles. The analysis by TG-DTG and SEM further showed that the dry–wet cycles led to the dissolution of struvite-K and the formation of more intrinsic micro-cracks.All specimens showed volume expansion in the full soak and dry–wet cycle test in the Na_2_SO_4_ solution and water environments; the final volume expansion value of the MKPC specimens under four corrosion conditions was in the order DW-II, M0 > SK-II, M0 > DW-II, M1 > SK-I, M0 > DW-I, M0. Under the SK-II conditions, the M0 paste with the highest strength revealed a higher volume expansion. The new needle-like hydrate crystals formed through the dissolution–precipitation of struvite-K may account for the volume expansion.Under the same number of dry–wet cycles, the strength test and volume stability test on the M1 paste confirmed that waterglass could effectively increase the durability under dry–wet cycling in the Na_2_SO_4_ solution. The micro analysis validated that waterglass can improve the compactness of the microstructure of MPC and prevent the dissolution of struvite-K.

## Figures and Tables

**Figure 1 materials-16-01101-f001:**
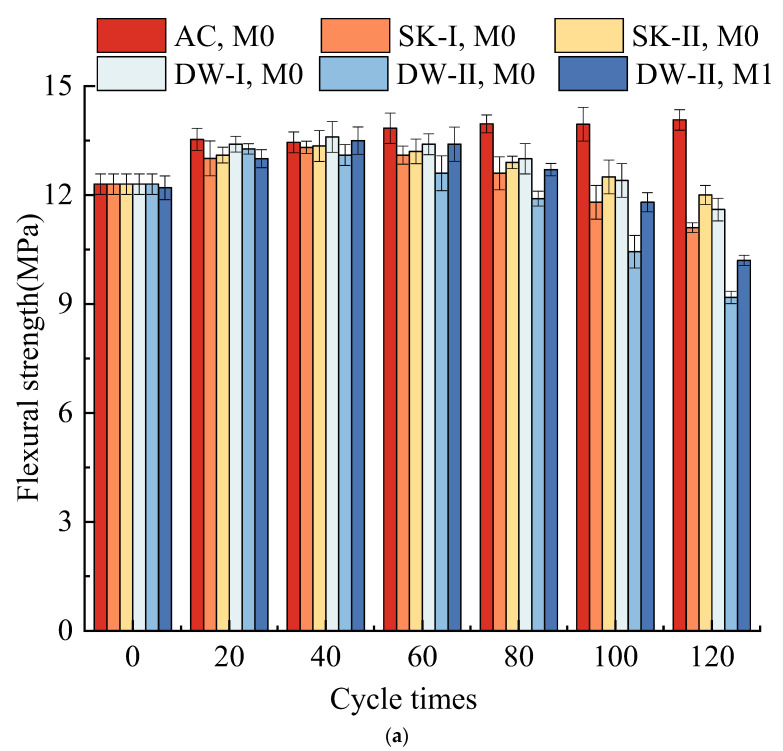
Flexural strength (**a**) and compressive strength (**b**) of MKPC mortars under different curing conditions.

**Figure 2 materials-16-01101-f002:**
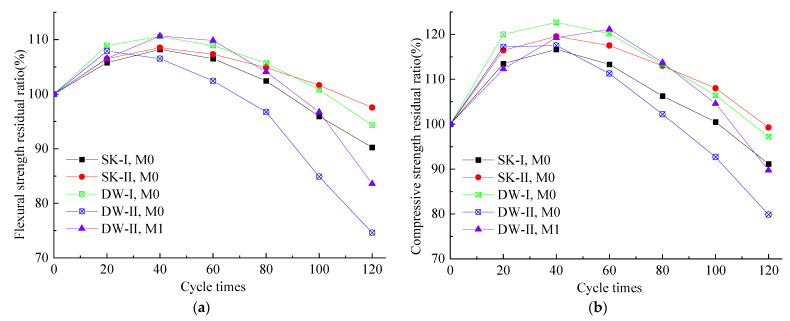
Flexural strength retention ratio (**a**) and compressive strength retention ratio (**b**) of MKPC mortars under different curing conditions.

**Figure 3 materials-16-01101-f003:**
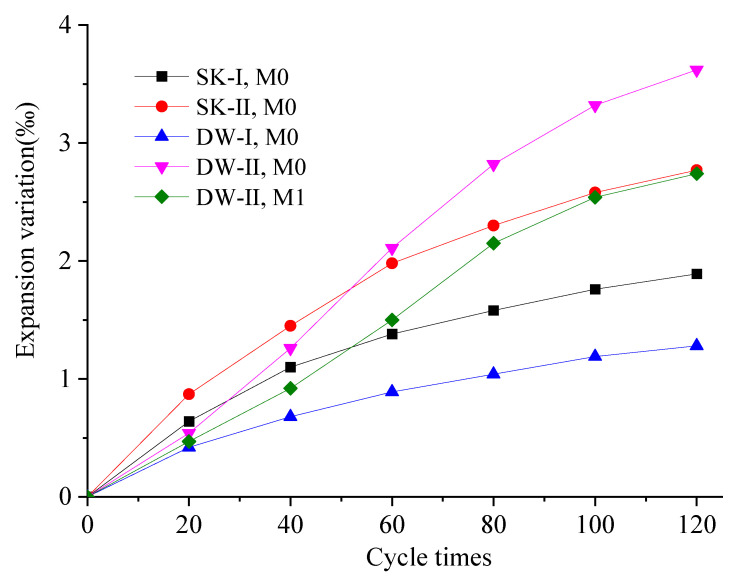
Deformation of the MKPC specimens under different corrosion conditions.

**Figure 4 materials-16-01101-f004:**
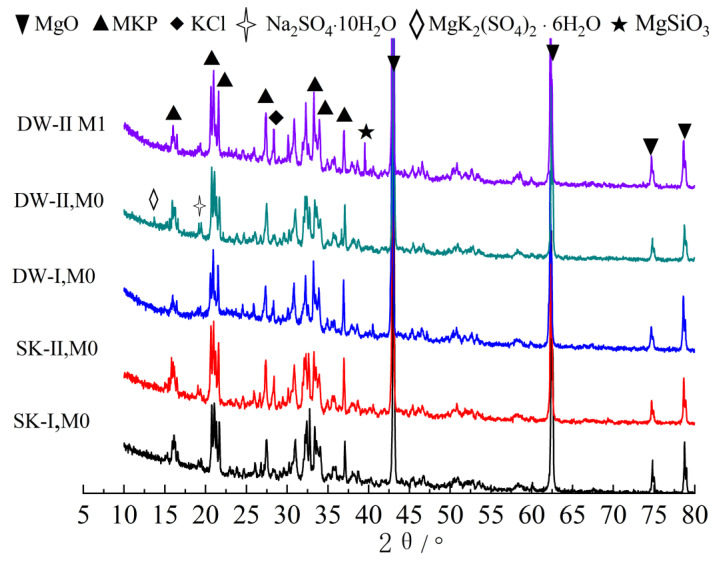
XRD curves of the MKPC samples under different corrosion conditions.

**Figure 5 materials-16-01101-f005:**
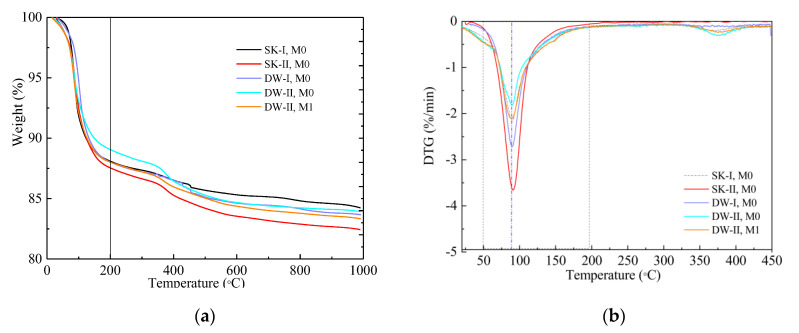
TG-DTG curves of the MKPC samples under different corrosion conditions. (**a**) TG, (**b**) DTG.

**Figure 6 materials-16-01101-f006:**
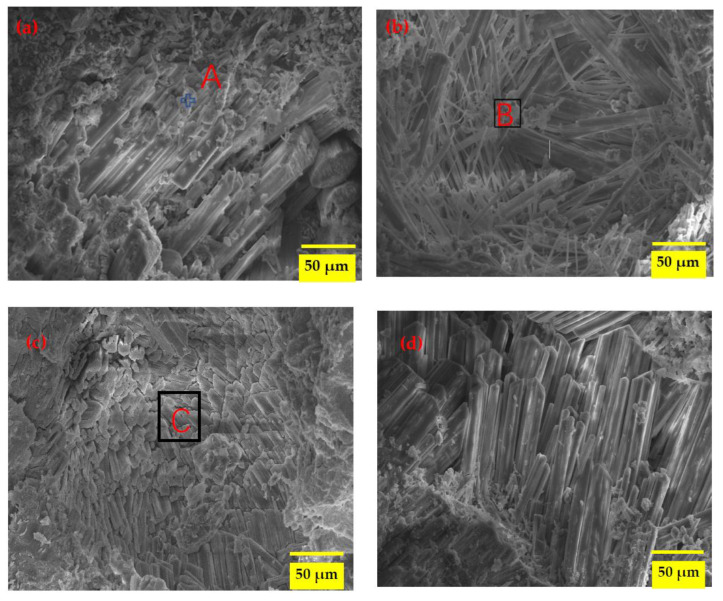
SEM photos of the MKPC samples under different corrosion conditions. (**a**) SK-I, M0, (**b**) SK-II, M0, (**c**) SK-II, M0, (**d**) DW-I, M0, (**e**) DW-II, M0, (**f**) DW-II, M1.

**Table 1 materials-16-01101-t001:** Chemical composition of the used MgO powder (wt.%).

Component	MgO	CaO	SiO_2_	Al_2_O_3_	Fe_2_O_3_	Na_2_O	K_2_O	Others
Content/%	91.85	3.14	3.68	0.17	0.87	-	-	0.30

**Table 2 materials-16-01101-t002:** Mixture proportions, fluidity, and initial setting time of the MKPC pastes.

Samples	Solid Materials			Liquid Materials		Fluidity (mm)	Initial Setting Time (min)
	MgO	PDP	Retarder (CR/MgO)	Waterglass (WG/MgO)	Water (*w*/*s* Ratio)		
M0	3.0	1	12%	0	0.115	162	20.5
M1	3.0	0.12	12%	2%	0.115	165	22.0

**Table 3 materials-16-01101-t003:** Details of the exposure conditions.

Abbreviation	Fresh Solutions	External Environment	Exposure Period
SK-I	tap water	Full soaking, 20 ± 5 °C	360 days
SK-II	5.0 wt.% Na_2_SO_4_ solution	Full soaking, 20 ± 5 °C	360 days
DW-I	tap water	Dry–wet cycles, 20 ± 5 °C	360 days (120 cycles)
DW-II	5.0 wt.% Na_2_SO_4_ solution	Dry–wet cycles, 20 ± 5 °C	

**Table 4 materials-16-01101-t004:** EDS results of different areas and points in hardened MKPC pastes.

Element	Atomic Percentage				
	Point A	Area B	Area C	Area D	Area E
O	60.64	55.12	70.75	68.79	55.8
Na	0.72	2.12	3.25	5.43	7.36
Mg	13.39	16.62	16.43	6.71	12.12
Si	-	-	-	-	1.60
P	13.23	12.34	3.54	7.05	11.07
K	11.90	13.15	3.52	6.81	6.75
S	-	0.04	2.18	4.37	5.30
CI	0.12	0.61	0.23	0.87	-

## Data Availability

Data are contained within the article.

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
