# Peer review of "Corrosion Behavior of Magnesium Potassium Phosphate Cement under Wet–Dry Cycle and Sulfate Attack"

_materials, 2023, doi:10.3390/ma16031101_

Round 1

Reviewer 1 Report

The paper "Corrosion behavior of magnesium potassium phosphate cement under Wet-Dry Cycle and sulfate attacks" presents a relevant theme and within the scope of this journal, and can be considered after some corrections suggested below:

(a) The abstract is generally well written, however in terms of content it is generic, i.e., the authors lack an in-depth study of the quantitative results of this research;

(b) Scientific innovation is limited in the introduction of the paper, the authors must go deeper and detail what this research differs from countless others that exist on this topic, this must be evidenced together with the objectives at the end of the introduction;

(c) The state of the art of the evaluated topic needs to be improved by the authors, note that some topics are absent and need to be known with current research, such as: 10.1016/j.cscm.2022.e01467; 10.1016/j.cscm.2022.e01264; 10.1016/j.cscm.2022.e01650.

(d) Is Table 1 presented correctly? Is there no information missing?

(e) Other information about the proportion of mixtures should be added in Table 2;

(f) “tap water” better explains this condition of the water used;

(g) The description of the experimental procedures is very short and unclear to readers;

(h) Discussions are very segmented in general, students should better detail these issues;

(i) The conclusion is too long, note this and make it leaner.

Reviewer 2 Report

This is a significant paper investigates the influence of dry-wet cycles and sulfate attack on the performance of concrete prepared with MKPC. The topic well-suits for the journal. The idea, the work done, and the results are excellent and novelty, the sentences and phrases in the article are comprehensible and the quality of English language appears good. I recommend major revision for the manuscript before final acceptance.

Following are my comments to improve the paper quality:

1.     The abstract should be a stand-alone section, therefore, all abbreviations in the abstract must be defined the first time you used them.

2.     Table 1 shows that the MgO used to prepare the MKPC is 91.85% pure, what if MgO with more impurities used?

3.     In the experimental section, you listed many standards considered for testing your samples, please cite these standards and add them to the reference list.

4.     In Figure 6, the letters in red and the squares in black should be removed or add explanation of what are these, it is not clear why you need these letters.

5.     For mechanical testing, loading rates of 40 to 60 N/s were considered for flexural strength measurement, this is a wide range which will affect your measurement for the different samples, how can you compensate for the difference in loading rate and it effect on the results? And how to ensure that your comparison is fair. Same for the compressive strength testing with loading rate of 2200 to 2600 N/s.

6.     The conclusion section is very long, please consider summarizing it. This part should only have a summary of the outcomes without going into detailed explanation of the results you obtained.

7.     I'd recommend spelling, grammar, and structure review for the whole manuscript for an easier understanding for readers.

8.     Please add more recent publications (no cited references form 2021 and only few from 2018, 2019, and 2020), here are some recommendations for you, I suggest you consider including them in your revised manuscript:

  • Improved durability of Saudi Class G oil-well cement sheath in CO2 rich environments using olive waste. Construction and Building Materials 262. https://doi.org/10.1016/j.conbuildmat.2020.120623.
  • Recent Advances in Magnesia Blended Cement Studies for Geotechnical Well Construction—A Review. Frontiers in Materials. https://doi.org/10.3389/fmats.2021.754431.

Round 2

Reviewer 1 Report

The authors have made all the indicated corrections.